# Domestic chickens activate a piRNA defense against avian leukosis virus

Yu Huining Sun[1], Li Huitong Xie[1], Xiaoyu Zhuo[2], Qiang Chen[1], Dalia Ghoneim[1], Bin Zhang[3], Jarra Jagne[4], Chengbo Yang[5], Xin Zhiguo Li[1]*

[1]Center for RNA Biology: From Genome to Therapeutics, Department of Biochemistry and Biophysics, Department of Urology, University of Rochester Medical Center, Rochester, United States; [2]Department of Genetics, Center for Genome Sciences and Systems Biology, Washington University School of Medicine, St. Louis, United States; [3]Department of Pathology and Laboratory Medicine, Department of Pediatrics, University of Rochester Medical Center, Rochester, United States; [4]Animal Health Diagnostic Center, Cornell University College of Veterinary Medicine, Ithaca, United States; [5]Department of Animal Science, University of Manitoba, Winnipeg, Canada

**Abstract** PIWI-interacting RNAs (piRNAs) protect the germ line by targeting transposable elements (TEs) through the base-pair complementarity. We do not know how piRNAs co-evolve with TEs in chickens. Here we reported that all active TEs in the chicken germ line are targeted by piRNAs, and as TEs lose their activity, the corresponding piRNAs erode away. We observed de novo piRNA birth as host responds to a recent retroviral invasion. Avian leukosis virus (ALV) has endogenized prior to chicken domestication, remains infectious, and threatens poultry industry. Domestic fowl produce piRNAs targeting ALV from one ALV provirus that was known to render its host ALV resistant. This proviral locus does not produce piRNAs in undomesticated wild chickens. Our findings uncover rapid piRNA evolution reflecting contemporary TE activity, identify a new piRNA acquisition modality by activating a pre-existing genomic locus, and extend piRNA defense roles to include the period when endogenous retroviruses are still infectious.

*For correspondence: Xin_Li@URMC.Rochester.edu

**Competing interests:** The authors declare that no competing interests exist.

## Introduction

A vertebrate germ-line genome faces repeated activation of transposable elements (TEs) as well as integration of new retroviruses that become endogenized. The genome has a marvelous ability to adapt to these challenges (*McClintock, 1984*). Among the adaptive responses, PIWI-interacting RNAs (piRNAs) are essential to protect the integrity of the germ-line genome by targeting 'non-self' sequences through base-pair complementarity. Disruption of piRNA pathways activates TEs in male and female fruit flies (*Wilson et al., 1996*; *Lin and Spradling, 1997*), male and female zebrafish (*Houwing et al., 2008*), and male mice (*Kuramochi-Miyagawa et al., 2004*; *Carmell et al., 2007*). piRNAs bind a specialized sub-family of Argonaute proteins, the PIWI proteins, which are expressed mainly in germ cells (*Kumar and Carmichael, 1998*; *Aravin and Hannon, 2008*; *Farazi et al., 2008*; *Kim et al., 2009*; *Thomson and Lin, 2009*; *Cenik and Zamore, 2011*). piRNAs guide PIWI proteins to their complementary RNA targets. PIWI proteins catalyze an endonucleolytic cleavage between the 10th and 11th positions of the RNA target relative to the piRNA 5′end. The cleaved product can then be loaded into another PIWI protein becoming a secondary piRNA. This results in a 'Ping-Pong' loop that amplifies antisense TE piRNAs (*Brennecke et al., 2007*; *Gunawardane et al., 2007*). The initial triggers of Ping-Pong amplification are produced from discrete genomic loci. The host needs to incorporate the foreign sequences into these piRNA-producing loci to recognize novel 'nonself'

**eLife digest** Viruses called retroviruses can infect animal cells and merge their genetic information with those of the animal causing damage to the animal's genetic blueprints. Once retroviruses are integrated into a cell they can sometimes get passed down through the generations over the centuries. Almost half of the human genetic code, for example, is made from ancient retroviruses and other foreign sequences. Over time many of these ancient viruses lost the ability to infect other cells and became trapped within cells but they can still jump out and damage the animal's genetic code under certain circumstances. These trapped foreign sequences are called transposable elements.

Animal cells produce molecules called piRNAs to shut down transposable elements. Most piRNAs are produced from genetic information that originally came from integrated retroviruses and that has been hijacked to defend the cell, a similar strategy as Crisper system in bacteria. Domestic chickens produce piRNAs against a virus called avian leukosis virus (or ALV for short) – which commonly infects domestic fowl. The virus also infected the wild ancestors of chickens, known as red jungle fowl, but these birds do not produce piRNAs. This provides an ideal setting to study the evolution of piRNAs in an animal that is not too distantly related to humans (chickens and humans both have backbones, and are therefore both warm-blooded vertebrates).

Sun et al. examined cells from the testicles of domestic chickens and red jungle fowl as an example of the role of piRNAs in protecting genetic information in vertebrates. The investigation revealed that piRNAs against all previously trapped viruses in the chicken's genetic code are produced in chickens to stop them from causing more damage. Sun et al. also observed the creation of piRNAs in chickens in response to ALV that had not yet become trapped in the chicken's genetic code. Importantly, the piRNAs could control these retroviruses while they were still infectious.

The experiments also revealed that piRNAs against ALV are produced from a single copy of ALV that is found in both domestic and wild chickens. The results showed that cells can produce new piRNAs using these pre-existing viral copies within their own genetics. This illustrates that production of piRNA from existing genetic material can be activated in response to certain cues.

Further work will seek to discover how existing genetic information becomes a source of piRNAs. In the United States, 8 billion domestic chickens are consumed each year, and a better understanding of how these birds defend themselves against viral infections could increase the productivity of the poultry industry around the world. Moreover, because other viruses trapped in the chicken's genetic code are related to similar viruses in humans, future discoveries made in this area could help to guide research that will benefit human health as well.

sequences, an RNA-based immune system similar to CRISPRs in prokaryotes (*Kumar and Chen, 2012*). New piRNA-producing loci can originate by duplication (*Assis and Kondrashov, 2009*), but duplication per se does not directly generate new piRNA sequences. The only known mechanism of new piRNA acquisition comes from studies of fruit flies, in which a TE inserts into an actively expressed piRNA cluster (*Khurana et al., 2011*). However, considering that active piRNA-producing loci represent only a tiny fraction of the genome, and that no preference for insertions of TEs into piRNA-producing loci has been reported (*Kumar and Chen, 2012*), other piRNA acquisition mechanisms remain to be discovered.

Endogenous retroviruses (ERVs) are distributed relative strictly in vertebrate genomes (*Gifford and Tristem, 2003*; *Eickbush and Jamburuthugoda, 2008*). Numerous distinct ERV families have invaded the chicken germ line (*Jurka et al., 2005*), making chicken (*Gallus gallus*) an excellent model to study virus-host interplay. Compared to other TEs in the chicken genome, including the ancient CR1 superfamily (*Vandergon and Reitman, 1994*) and DNA transposons, and largely absent short interspersed nuclear elements (SINEs) (*International Chicken Genome Sequencing Consortium, 2004*), chicken ERVs are more active and have led to phenotypic changes like blue egg-shells (*Wang et al., 2013*) and late feathering (*Boyce-Jacino et al., 1989*). Chicken ERVs can also remain infectious, and may evolve into new viruses through recombination with host genes or exogenous viruses. Avian leukosis virus (ALV) was the first ERV to be discovered (*Temin, 1964*;

*Weiss, 1969*; *Baluda, 1972*). Uninfected chickens sometimes spontaneously shed infectious ALV subtype E (ALVE) viruses (*Varmus et al., 1972*; *Weiss, 2006*), and the infection can lead to cancer (*Weiss, 2006*). ALV acquisition of a host oncogene, *SRC*, generated a more acute transforming virus, Rous sarcoma virus (RSV) (*Stehelin et al., 1976*). Recombination between ALVE and EAV-HP—a member of the endogenous avian retrovirus (EAV) family, has created ALV subtype J (ALVJ), a new subgroup of ALVs that were associated with myeloid leukosis in meat-type chickens (*Smith et al., 1999*). After spreading to China in 2002, ALVJ mutated to infect egg-laying breeds with a wide spectrum of tumors (*Gao et al., 2010*). Despite the threat of ERVs to the poultry industry, we lack a systematic investigation of ERV activity and piRNA-mediated suppression in the chicken germ line.

Here, we identified the ERV activity and piRNA-producing loci in White Leghorn, the most popular egg-laying breed, dissected the interplay between ERVs and piRNAs, and traced the origination of recently acquired piRNAs in undomesticated wild chickens, Red Jungle Fowl (*Eriksson et al., 2008*). We chose to focus on the White Leghorn because this domestic breed has suffered from ERV activation and thus its ERVs have been extensively studied (*Crittenden, 1991*). White Leghorn lays an average of 280 eggs per year (*Bao et al., 2008*). Breeders have used the late-feathering trait as a convenient marker to select female layers at hatch (*Boyce-Jacino et al., 1989*); however, this trait is linked to a fully infectious ALVE provirus (known as *ev21*) on the Z chromosome, which causes decreased performance (*Smith and Fadly, 1988*; *Fadly and Smith, 1997*). We found that 73 TE families, including ALVE, were active in White Leghorn testis, and all 503 TE insertions absent in Red Jungle Fowl were derived from these TE families. More than 60% of the active TEs belonged to ERV families, indicating that ERVs contribute to most TE activity in chickens. All active TEs are targeted by robust piRNA-mediated suppression. As TEs become inactivated, their targeting piRNAs erode away. We found that the ability of chickens to produce piRNAs targeting ALV is an evolutionarily recent acquisition—White Leghorn produced abundant ALV piRNAs while Red Jungle Fowl did not. The ALV piRNAs in White Leghorn were produced from a truncated ALV provirus that was known to render its host ALV resistant (*Robinson et al., 1981*). The presence of this provirus predated domestication, indicating that the responsible genomic region exists in either an 'on' or 'off' state as a piRNA-producing locus.

## Results

### Identifying active TEs in domestic fowl

Active ERVs are transcribed and translated, and are able to transpose within the germ line. Comparison of RNA-seq data from 12 tissues (*Brawand et al., 2011*) indicated that chicken ERV families were ubiquitously expressed (*Figure 1—figure supplement 1A*). Because many insertions are truncated, detection of ERV RNAs does not necessarily indicate that they are translated or competent for transposition. It has been shown that chicken ERVs are transcribed and translated in somatic cells (*Bolisetty et al., 2012*). Because the highest expression was in testis and ovary—the only tissues where their expansion can become heritable (*Figure 1—figure supplement 1A*), we decided to perform polysome profile analysis to determine whether ERVs were also being translated in testis. Adult White Leghorn testis lysates were separated in 10–50% sucrose-density gradients by ultracentrifugation (*Figure 1A*). This fractionation separates non-translating ribonucleoproteins, small and large subunits of ribosomes, monosomes and polysomes, as shown by the distribution of rRNA. Actively translated *β-ACTIN, CILI,* and *CIWI* mRNAs co-sedimented with both monosome and polysome fractions, but the *MALAT1* non-coding RNA did not co-sedimented with polysomes. CILI and CIWI are the two PIWI proteins in chickens (*Figure 1—figure supplement 2*). We tested the distribution of *CR1-B* and *CR1-F* families that belong to the CR1 superfamily, as well as the *EAV-HP* and *ALVE* that belong to ERV families. Although CR1 arose prior to the divergence of birds and reptiles and peaked ~45 million years ago (*Vandergon and Reitman, 1994*), CR1-F and CR1-B elements remain able to drive their own transcription (*Wicker et al., 2005*; *Lee et al., 2009*). *CR1-B*, *CR1-F*, *EAV-HP*, and *ALVE* transcripts co-sedimented with polysomes. These profiling results suggest that *CR1-B*, *CR1-F*, *EAV* and *ALVE* insertions are transcribed and translated in testis.

To determine whether the observed co-sedimentation with ribosomes reflects the active translation, we performed ribosome profiling using testis lysates from adult White Leghorn. Ribosome profiling is based on the facts that the ribosome-bound fraction of mRNA is protected from RNase

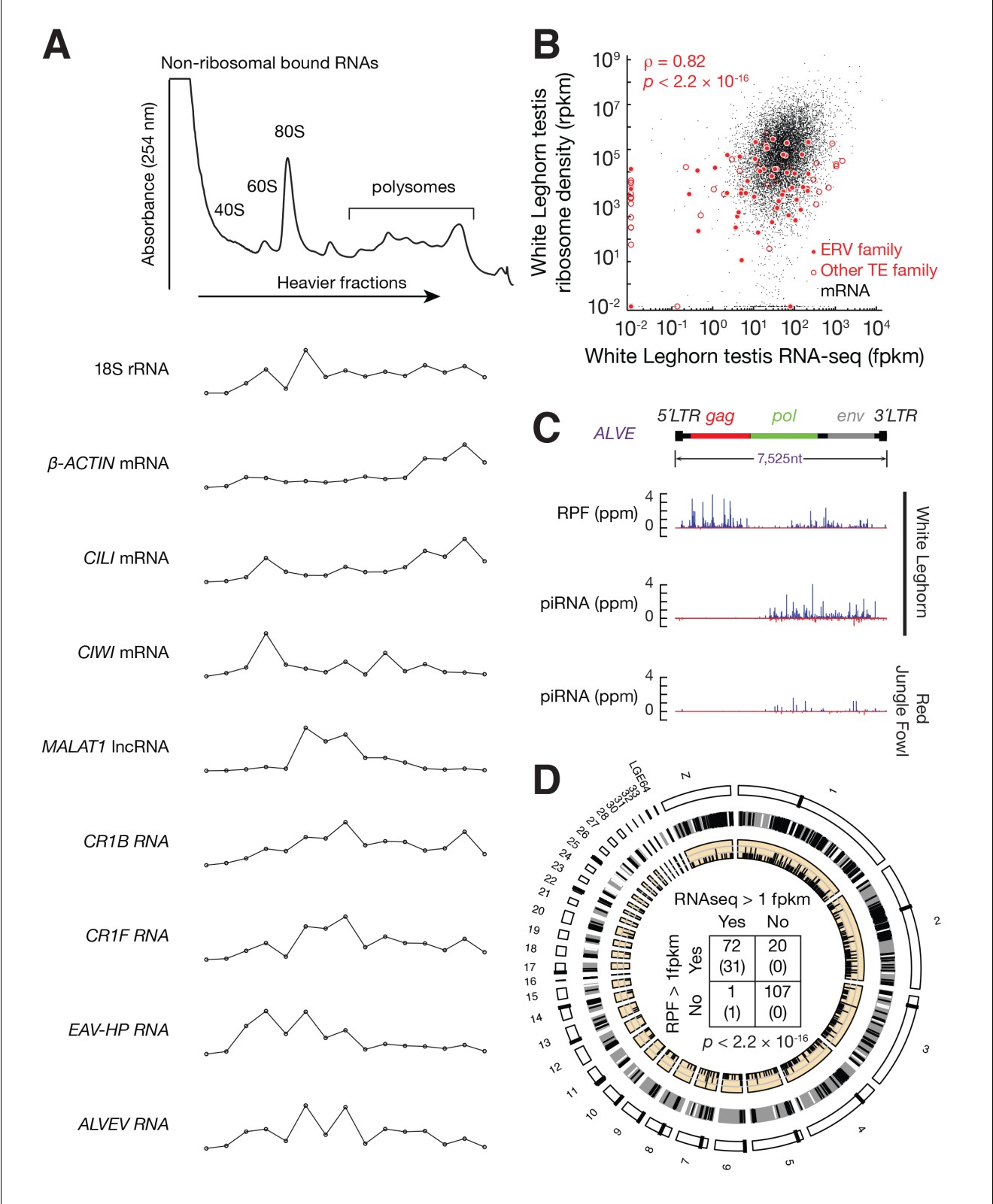

**Figure 1.** Active ERVs in White Leghorn testis. (**A**) A254 absorbance profile of 10% to 50% sucrose density gradients of testis lysates from adult rooster. From top to bottom, plots show the relative abundance of 18S rRNA, *β-ACTIN* mRNA, *CILI* mRNA, *CIWI* mRNA, chicken *Malat1* lncRNA, *CR1B*, *CR1F*, *EAV-HP*, and *ALVE* quantified by RT-qPCR. Data were normalized to a spike-in control RNA. (**B**) Scatter plots of transcript abundance versus ribosome density. Each black dot represents an mRNA expressed in testis. Each filled red circle represents an ERV family, and each open red circle represents

*Figure 1 continued*

any other TE family, including DNA transposons and CR1 superfamily; rpkm, reads per kilobase of transcript per million mapped reads; fpkm, fragments per kilobase of transcript per million mapped reads. (**C**) Normalized reads of White Leghorn RPFs (Top), White Leghorn piRNAs (Middle), and Red Jungle Fowl small RNA reads (>23 nt) (Bottom). Blue represents sense mapping reads; Red represents anti-sense mapping reads. The gene organization of ALVE is also shown. *Gag*, group-specific antigen; *Pol*, polymerase; *Env*, envelope protein; ppm, parts per million. (**D**) Circos plot representing the locations, from periphery to center, of cytological position (black lines represent centromeres), piRNA clusters in White Leghorn (Black lines represent conserved piRNA clusters; White lines represent divergent piRNA clusters), putative new insertions discovered by TEMP (tiles) using genomic resequencing of White Leghorn, and 2 × 2 contingency table for Fisher's exact test to assess the significance of the coincidence of transcription and translation of each TE family. The table data correspond to the number of TE families in each category and, in parentheses, the number of TE families in each category with recent transpositions.

The following figure supplements are available for figure 1:

**Figure supplement 1.** Tissue distribution of ERVs and piRNA pathway genes.

**Figure supplement 2.** PIWI proteins are conserved between mammals and birds.

**Figure supplement 3.** Ribosome profiling in adult rooster testes.

**Figure supplement 4.** A recent ALVE insertion in the *SOX5* gene detected by genome-resequencing of White Leghorn.

digestion in vitro (*Steitz, 1969*), and that the subsequent genome-wide sequencing of ribosome-protected fragments (RPFs) provides a snapshot of in vivo translation (*Ingolia et al., 2009*). RNA fragments protected from RNase A and T1 digestion were isolated from 80S fractions and sequenced (*Figure 1—figure supplement 3A*) (*Ricci et al., 2014*; *Cenik et al., 2015*). Similar to reported RPF sizes in mammals, the RPF sizes in chicken from coding DNA sequences (CDS) ranged from 26–32 nt (*Figure 1—figure supplement 3B*). While RNA-seq reads were distributed throughout the entire set of mRNA transcripts, RPF reads were enriched in CDS regions (*Figure 1—figure supplement 3C*), and RPFs that mapped to CDS regions accounted for 96% of the RPFs that mapped to entire mRNA transcripts. The RPF reads mapping to open read frames displayed an obvious three-nt periodicity (*Figure 1—figure supplement 3D*), reflecting the triplet nature of the genetic code during translation elongation. Based on the enrichment of RPFs at CDS regions and the observed codon periodicity of RPFs, we conclude that the ribosome profiling identified RNAs undergoing translation.

We integrated the ribosome profiling and RNA-seq data in our analysis of transcription and translation of ERVs and other TE families. The ribosome density of each TE family correlated with their steady-state RNA levels ($\rho = 0.82$, $p<2.2\times10^{-16}$) (*Figure 1B*). The median translational efficiency (ratio of ribosome density to transcript abundance) of ERVs was around 1/10 of the median translation efficiency of mRNAs that were expressed in testis. Consistent with our expectation that ALVE was active in White Leghorn, we detected RPFs mapping to ALVE (*Figure 1C*). These ALVE RPF reads displayed a length distribution that was similar to that of RPFs from CDS regions (*Figure 1—figure supplement 3B*), and they displayed codon periodicity (*Figure 1—figure supplement 3D*). These RPFs were distributed throughout the entire ALVE transcripts but with higher abundance at *gag* and *env* than at *pol* (*Figure 1C*). Most transcribed TEs were also translated (the two events, transcription and translation, were significantly associated: Fisher's exact test, $p<2.2\times10^{-16}$; *Figure 1D*). We detected RPFs in 71 of 73 TE families that were transcribed (97.3%). Sometimes RPFs could not be unambiguously assigned to TEs due to their small sizes, resulting in false positive signals on transcriptionally silenced TEs. Based on RNA-seq data, polysome profiles, and ribosome profiling, we conclude that most transcribed TE families in the testis were also being translated.

To detect new transposition events, we aligned the published resequencing data of the White Leghorn genome with ~100X coverage (*Oh et al., 2016*) to the Red Jungle Fowl reference genome. 503 putative TE insertions, absent in Red Jungle Fowl, were distributed throughout the chicken genome (*Figure 1D*, *Supplementary file 1*). Although Red Jungle Fowl are commonly called as the 'ancestor' of domestic chicken, they evolved thousands of years in parallel with domestic chicken after chicken domestication (*West and Zhou, 1988*), therefore our analysis cannot distinct lineage

specific insertions in White Leghorn from lineage specific deletions in Red Jungle Fowl using structure variant alone. De novo ALVE insertions have been reported in domestic fowl (*Crittenden, 1991*), and we detected new ALVE insertions in *SOX5*, as recently reported (*Rutherford et al., 2016*). The identification of a 6 bp target site duplication typical for ALVE (*Figure 1—figure supplement 4*) indicates a transposition event rather than a genome duplication event. No putative new insertion came from the 127 transcriptionally inactive TE families (*Figure 1D*), confirming their inactive state. All putative new insertions came from the 73 actively transcribed TE families. Thus, combing multiple methods to detect active ERVs, we found that active TEs and their insertions in the White Leghorn genome (*Supplementary file 1*). Our data indicated that transposing activity of some TEs has been recent or may have been ongoing.

## Chicken piRNAs reflect contemporary TE activity

Active TEs must be tightly controlled in germ cells. Using small RNA-seq data from the adult testis of White Leghorn (*Li et al., 2013*), we detected abundant TE piRNAs, which accounted for 7.8% of total piRNAs, and exhibited a size range peaking at 24–25 nt. These small RNAs were resistant to oxidation (*Figure 2—figure supplement 1A,B*). Oxidation by sodium periodate makes most small RNA species non-accessible for cloning into libraries, but 2′-O-methyl-modified 3′ termini protect piRNAs from oxidation (*Ghildiyal et al., 2008*). Like piRNAs in other species, these TE piRNAs typically began with uracil (61.6% of species and 66.7% of reads, *Figure 2—figure supplement 1C*). Almost equal numbers of piRNAs mapped to sense versus antisense strands (median ratio of sense to antisense piRNAs was 1.2) (*Figure 2—figure supplement 1D*), and there was an adenine bias at the 10th position (*Figure 2—figure supplement 1C*), indicating that secondary piRNAs are generated (*Brennecke et al., 2007*; *Gunawardane et al., 2007*). To test whether the anti-sense TE piRNAs were able to guide the PIWI proteins to cleave TE transcripts, we plotted the distance between the 5′ends of anti-sense piRNAs and the 5′ends of sense piRNAs from TE loci. We detected a significant Z score at a distance of 10 nt, a signature of robust Ping-Pong amplification (*Figure 2—figure supplement 1E*) (*Brennecke et al., 2007*; *Gunawardane et al., 2007*). These findings indicate that a piRNA mediated silencing pathway against TEs is active in the chicken germ line.

The expression of piRNAs that target each TE family correlated with overall TE expression ($\rho = 0.81$, $p < 2.2 \times 10^{-16}$, *Figure 2A*; the two events were significantly associated: Fisher's exact test, $p < 2.2 \times 10^{-16}$, *Figure 2B*), although there were exceptions. All the 73 actively expressed TE families were targeted by piRNAs. The presence of this piRNA activity explains why expression of the TEs can be tolerated in White Leghorn. Most inactive TEs (108 families) are not targeted by piRNAs; interestingly, 19 inactive TEs are targeted by piRNAs. Those piRNAs that target inactive TEs exhibit the authentic piRNA length distribution, resistance to oxidation, and first position (1st) U bias (*Figure 2—figure supplement 2A,B*), but they are less abundant than the piRNAs that target active TEs ($p < 2.2 \times 10^{-16}$) (*Figure 2C*). piRNAs that target active TEs display a robust Ping-Pong amplification with a median Z score of 12.2; whereas, the piRNAs that target the inactive TEs do not show significant Ping-Pong amplification (median Z score of 0.41) (*Figure 2C*; Z-score >3.3 corresponds to p<0.01), although both sense and antisense TE piRNAs are produced in equal abundance (*Figure 2—figure supplement 1B*). The lack of a Ping-Pong signature for piRNAs targeting inactive TEs supports the function of Ping-Pong amplification as an adaptive response to TE activation rather than merely a consequence of piRNA production.

Since inactive TEs are no longer a threat to the host genome, piRNAs that target them could either represent a remnant from suppression of past threats, or have acquired new functions beyond TE suppression. To distinguish these two possibilities, we grouped TEs based on their expression and based on TE piRNA expression (*Figure 2A,B*), and compared TE age (*Figure 2D*). If inactive TEs that were targeted by piRNAs have an intermediate age between active TEs and inactive TEs that were not targeted by piRNAs, the targeting more likely reflects a remnant of prior suppression function; if inactive TEs that were targeted by piRNAs are as old as other inactive TEs, it is more likely that these TEs and piRNAs represent possible new functions. We inferred TE age using organism information available in Repbase (*Jurka et al., 2005*). We found that all active TEs that were targeted by piRNAs are recent invaders of the chicken genome. These young TEs are specific to *Gallus gallus* or other *Gallus* genera. ERVs comprised most of these young TEs (47 out of 73 families), which is consistent with the observations that non-ERV TEs lack recent activity in chickens (*International Chicken Genome Sequencing Consortium, 2004*). More than 90% of inactive TEs

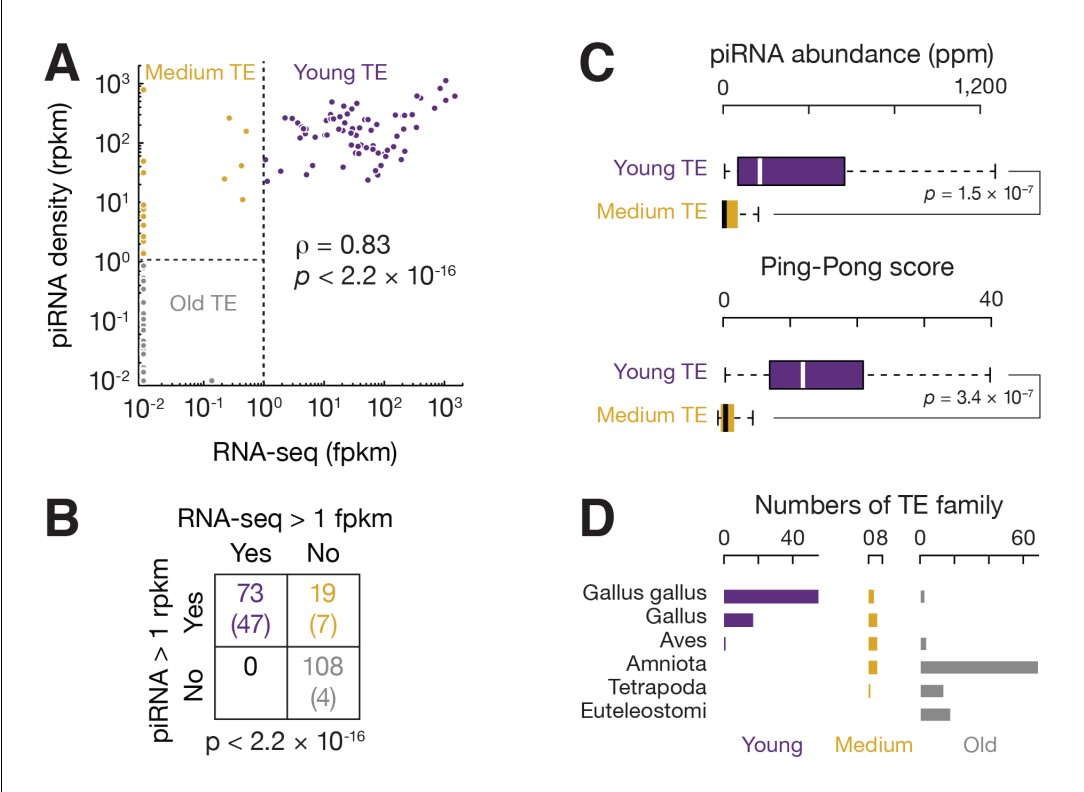

**Figure 2.** Three groups of TEs based on TE expression and piRNA expression. (**A**) Scatter plots of TE transcript abundance versus TE piRNA abundance. Each filled circle represents a TE family. Here and in *Figure 2—figure supplements 1* and *2*, young TE in purple, medium TE in yellow, and old TE in grey. (**B**) 2 × 2 contingency table for Fisher's exact test to assess the significance of the coincidence of the TE transcript abundance and TE piRNA abundance. The table data correspond to the number of TE families in each category and, in parentheses, the number of ERV families in each category. (**C**) Top, box plots present piRNA abundance per TE family. Bottom, box plots present Ping-Pong amplification score per TE family. (**D**) Histograms of TE ages.

The following figure supplements are available for figure 2:

**Figure supplement 1.** piRNA-mediated TE suppression in rooster testes.

**Figure supplement 2.** Medium TE piRNAs are authentic piRNAs.

that were not targeted by piRNAs had invaded the chicken genome before birds and other amniotes diverged. These old TEs included 80 DNA transposons, 15 CR1s, and 4 ERVs. We found that the 19 inactive TEs that were targeted by piRNAs were of medium age, exhibiting invasion times that were distinct from both old and young TEs (*Figure 2D*, $\chi^2$, p≤2.5×10$^{-9}$). From these data, we conclude that piRNA expression reflects TE age—young TEs are targeted by abundant piRNAs, while TE inactivation leads to the erosion of piRNA production. Thus, we designate three TE groups: young, medium, and old based on the expression pattern of TEs and TE piRNAs (*Figure 2A,B*). Our data imply that a rapid turnover of chicken piRNA sequences reflects contemporary TE activity.

## ALVE-targeting piRNAs are found in domestic but not wild chickens

piRNAs have not previously been shown to suppress any infectious virus—exogenous or endogenous—in vertebrates; yet, we unexpectedly detected abundant piRNAs mapping to ALVE in adult rooster testis (*Figure 3A*). These ALVE-mapping reads, which peaked at 25 nt were resistant to oxidation, indicating that they were authentic piRNAs. These piRNAs mapped to both strands of ALVE (*Figure 1C*), spanning *env* and the 3′ half of *pol*. The sense piRNAs exhibited a typical 1st U bias, and the antisense piRNAs exhibited a 10th A bias (*Figure 2—figure supplement 2C*), indicating

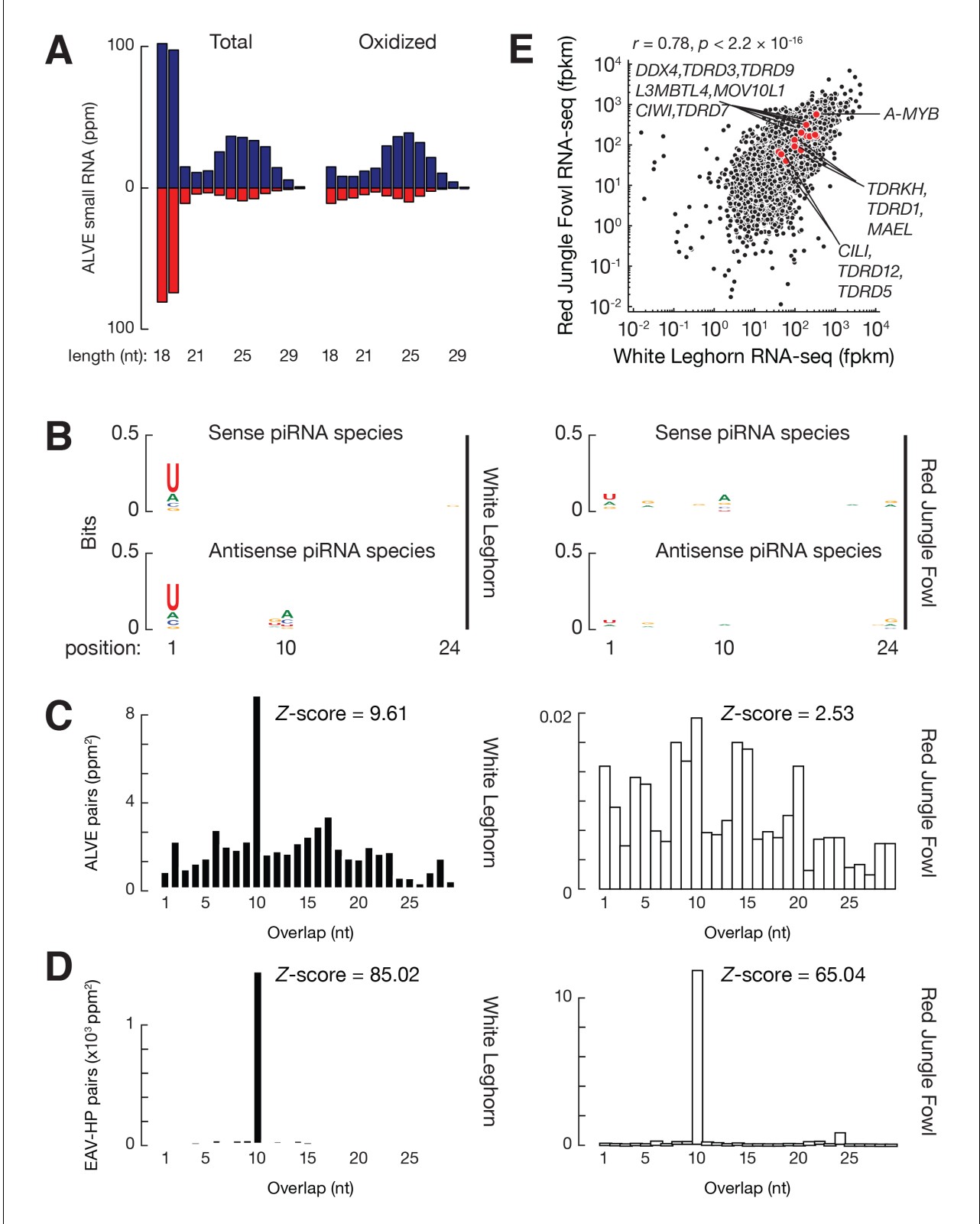

**Figure 3.** ALVE piRNA acquisition. (**A**) Length distributions of testis small RNAs mapping to ALVE. Blue represents sense mapping piRNAs; Red represents anti-sense mapping piRNAs. (**B**) Sequence logo showing the nucleotide composition of ALVE piRNA species from White Leghorn (*left*) and ALVE species from Red Jungle Fowl (*right*), top, sense ALVE mapping reads, bottom, anti-sense ALVE mapping reads. (**C**) Analysis of the 5′–5′ overlap between ALVE piRNAs from opposite strands was analyzed. Significance of ten-nucleotide overlap ('Ping-Pong') was determined from Z-score.
*Figure 3 continued on next page*

*Figure 3 continued*

Z-score >3.3 corresponds to p-value<0.01. (D) Analysis of the 5′−5′ overlaps between EAV-HP piRNAs from opposite strands. (E) Scatter plots comparing mRNA abundance between White Leghorn and Red Jungle Fowl. Each black filled circle represents an mRNA expressed in testis, and each red filled circle represents an mRNA coding for a protein in the piRNA pathway.

production of secondary piRNAs. Indeed, robust Ping-Pong amplification signals were detected in ALVE piRNAs (*Figure 3B*). ALVE piRNAs were produced to an abundance of 188 parts per million reads mapped to the genome (ppm), which was roughly half the abundance of EAV-HP piRNAs (359 ppm). The EAV family underwent endogenization prior to *Gallus* speciation, and although its members no longer produce viral particles, they actively transpose and can cause new insertions (*Boyce-Jacino et al., 1992*) as reported in *Supplementary file 1*. Because small RNAs recognize their targets without complete sequence complementarity (*Bartel, 2009*), the roughly 3000 ALVE piRNA species detected can recognize mutated ALVE, which thus might explain why other ALV family members failed to endogenize. The presence of these piRNAs likely improves fitness both by suppressing the mutagenic effects of germ-line activation, and by reducing the numbers of viral copies thereby reducing horizontal transmissions.

ALVE was introduced into the chicken genome following speciation but prior to domestication (*Frisby et al., 1979*). The Red Jungle Fowl genome carries one full length ALVE, ALVE-JFvB (*Weiss and Biggs, 1972*), and one truncated copy, ALVE6 (known as ALVE-JFvA in Red Jungle Fowl and *ev6* in White Leghorn) (*Levin et al., 1994*; *Benkel and Rutherford, 2014*). Using published RNA-seq data from the testis of Red Jungle Fowl (*Necsulea et al., 2014*), we determined the expression of ALVE to be 17.1 fragments per kilobase of transcript per million mapped reads (fpkm), which is roughly one-third of the abundance of EAV-HP (49.3 fpkm). Based on 41 single nucleotide polymorphisms (SNPs) that distinguish ALVE-JFevB and ALVE6, we determined that the two copies were expressed at a 1:5 ratio. Given the level of testicular expression of ALVE, it was surprising that we did not detect robust ALVE piRNAs in Red Jungle Fowl (*Figure 1C*). These ALVE-mapping reads in Red Jungle Fowl had neither a strong U bias (*Figure 3B*), nor significant Ping-Pong amplification (*Figure 3C*). The Ping-Pong analysis method based on the Z-score of piRNA pairs is not affected by sequencing depth (*Zhang et al., 2011*). Therefore, the absence of robust expression of ALVE piRNAs in Red Jungle Fowl indicates that the germ-line endogenization of a new retrovirus is not sufficient to establish piRNA-mediated repression.

To determine whether piRNAs are increased generally for all TEs or specifically for ALVE in White Leghorn, we tested the expression of EAV-HP piRNAs in Red Jungle Fowl. EAV-HP piRNAs exhibited robust Ping-Pong amplification in both White Leghorn and Red Jungle Fowl (*Figure 3D*). Moreover, compared to the non-TE piRNAs, the overall percentage of TE piRNAs did not increase in White Leghorn ($\chi^2$, p=1). We then compared expression of piRNA pathway genes in White Leghorn and Red Jungle Fowl (*Figure 3E*). RNA silencing pathway genes, including *CIWI, CILI, A-MYB* (*Li et al., 2013*), *DDX4* (*Kuramochi-Miyagawa et al., 2010*), *MAEL* (*Soper et al., 2008*), *L3MBTL4* (*Fagegaltier et al., 2016*; *Sumiyoshi et al., 2016*), *MOV10l1* (*Frost et al., 2010*; *Zheng et al., 2010*), *TDRD1* (*Chen et al., 2009*; *Kojima et al., 2009*; *Reuter et al., 2009*; *Wang et al., 2009*), *TDRKH (TDRD2)* (*Saxe et al., 2013*), *TDRD3, TDRD5* (*Yabuta et al., 2011*), *TDRD7* (*Tanaka et al., 2011*), *TDRD9* (*Aravin et al., 2009*; *Shoji et al., 2009*), and *TDRD12* (*Aravin et al., 2009*; *Shoji et al., 2009*), exhibited a median abundance of 164 fpkm in White Leghorn testis, which was not significantly different from expression in Red Jungle Fowl (median abundance of 167 fpkm, p=0.63). This expression of TE piRNAs and piRNA processing genes at approximately the same levels in Red Jungle Fowl and White Leghorn suggests that the activation of piRNAs in White Leghorn is specific to ALVE. The presence of an active piRNA pathway in Red Jungle Fowl indicates that ALVE piRNA expression emerged or was selected subsequent to domestication.

## Defining piRNA-producing loci in chickens

The ability of chickens to produce piRNAs against a new ERV provides a rare opportunity to study where new piRNAs are acquired. To identify the genomic source of the ALVE piRNAs, we defined all piRNA-producing loci, so-called piRNA clusters, in White Leghorn. Using our previously developed dynamic programming algorithm (*Li et al., 2013*), in total, we identified 1633 piRNA clusters that

accounted for 0.88% of the chicken genome, and explained 87.3% of total piRNA reads and 81.1% of uniquely mapping piRNAs (*Figure 4A*). These piRNA clusters were distributed on most autosomes and the Z chromosome (*Figure 1D*). Unlike divergently and uni-directionally transcribed mouse piRNA-producing loci, we observed that chicken piRNAs were produced from both strands of piRNA clusters (*Figure 4B*, *Figure 4—figure supplement 1A*) as reported previously (*Li et al., 2013*; *Chirn et al., 2015*), and were derived from convergently transcribed precursors detected by our RNA-seq data (*Figure 4B*, *Figure 4—figure supplement 1A*). Over 70% of clusters (1173 out of 1633) included uniquely mapping piRNAs transcribed from either strand at a level of greater than 10% of total uniquely mapping piRNAs from that cluster. Based on these findings, we conclude that most chicken piRNA-producing loci are transcribed from both strands, and both transcripts are processed into piRNAs.

We tested tissue-specific expression of piRNA precursors using the RNA-seq data to measure the abundance of piRNA cluster transcripts in testis as well as in 11 other tissues (both sexes were included) (*Figure 4C*). The median abundance of piRNA precursors in the testis was 2.6 fpkm, but we were unable to detect these transcripts in other tissues (median abundance = 0). The expression of two PIWI genes, *CILI* and *CIWI*, was also only detected in testis (*Figure 1—figure supplement 1B*). The detection of piRNA precursors and PIWI mRNAs exclusively in testis is consistent with their role in protecting the germ-line genome. The lack of detection of PIWI mRNAs and piRNA precursors in ovary suggests that chicken piRNA pathways display sexual dimorphism. In mouse and chicken, the synthesis of most piRNA precursors and mRNAs of key piRNA pathway genes is driven by the transcription factor A-MYB (*Li et al., 2013*). We found that *A-MYB* is also expressed exclusively in the testis, which may explain the transcriptional activation of piRNA pathways in chicken (*Figure 4C* and *Figure 1—figure supplement 1B*). Thus, the chicken piRNA pathway is transcriptionally activated predominantly in the testis.

## A previously defined ALV-resistance locus produces ALVE piRNAs

Based on our finding that the expression of ALVE piRNAs was acquired recently, we reasoned that the ALVE piRNAs were either derived from new ALVE insertions or activated from pre-existing genomic elements. None of the new ALVE insertions in White Leghorn were found within or near the 1633 identified piRNA-producing loci (*Supplementary file 1*). To assess the second possibility, we systematically compared piRNA cluster locations in White Leghorn and Red Jungle Fowl. Using the same parameters in the dynamic programming algorithm, we defined the piRNA clusters in Red Jungle Fowl. Overall, the genomic location of 72% piRNA-producing loci (1168 of 1633) overlapped between the two breeds, but 468 piRNA clusters are specific to White Leghorn (the right circle in *Figure 4D*). The piRNAs from divergent piRNA clusters exhibited authentic piRNA length distribution, resistance to oxidation, and 1st U bias (*Figure 4—figure supplement 1B,C*). In White Leghorn, the conserved piRNA clusters accounted for 77.4% of uniquely mapping piRNAs, and the divergent piRNA clusters only accounted for only 3.7% of uniquely mapping piRNAs. In Red Jungle Fowl, 82.8% of total piRNAs and 74.9% of uniquely mapping piRNAs (*Meunier et al., 2013*) could be explained by the conserved piRNA clusters (*Figure 4A*). Thus, piRNAs are predominantly produced from identical genomic locations but with notable divergence between the breeds.

One divergent piRNA-producing locus (cluster 719) contains the truncated ALVE provirus (*Figure 4E*), ALVE6. Cluster 719 produces abundant piRNAs (147 ppm) in White Leghorn, but in Red Jungle Fowl produces few piRNAs (*Figure 4E*). ALVE6 has lost its 5′LTR, *gag*, and half of *pol*, eliminating its transcriptional promoter (*Tereba, 1981*). The gene structure matches the distributions of piRNAs on ALVE, which starts in the middle of the *pol* gene (*Figure 1C*). This defective provirus has been associated with ALVE resistance (*Robinson et al., 1981*). The wide distribution of ALVE6 in commercial egg-laying breeds has been believed to reflect selection of nonshedders (*Hayward et al., 1980*; *Kuhnlein et al., 1989*; *Smith et al., 1990b*). ALVE6 is the only known ALVE provirus that is present in both White Leghorn and Red Jungle Fowl (*Levin et al., 1994*). In addition to the resequencing analysis of White Leghorn, we used the longer sequencing reads of Sanger sequencing to confirm that although sequence polymorphisms exist, the genomic structure of the ALVE6 locus surrounding regions was remains the same as the reference locus in Red Jungle Fowl (*Figure 4—figure supplement 2A*). These results indicate that ALVE6 existed in the chicken genome prior to domestication.

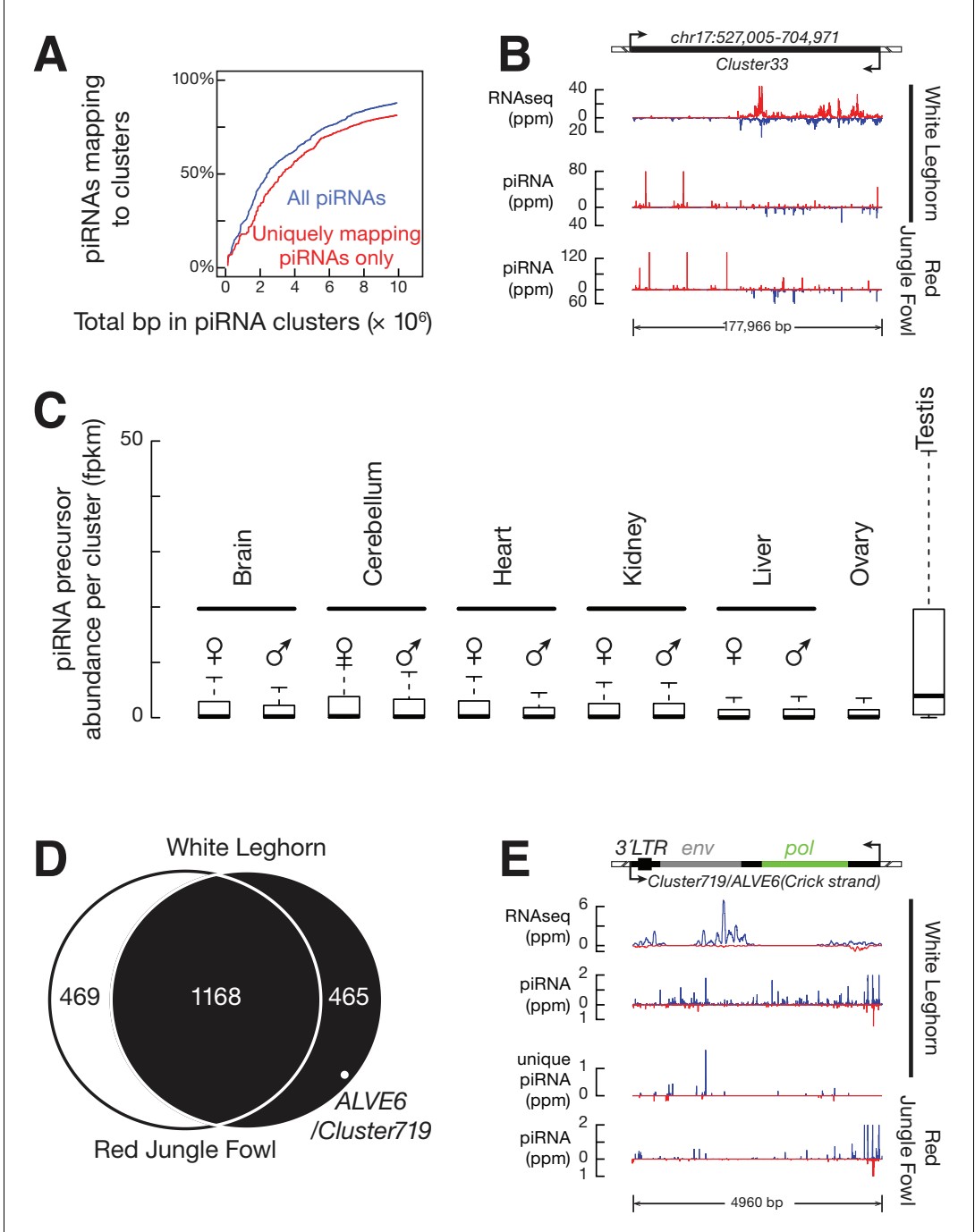

**Figure 4.** ALVE6 is the primary piRNA-producing locus for viral piRNAs. (**A**) Cumulative distributions for all piRNAs (Blue) and for uniquely mapping piRNAs (Red) on the 1633 piRNA loci in White Leghorn. (**B**) An example of conserved piRNA-producing loci, *Cluster33*, in chicken. Normalized RNA-seq reads of piRNA precursors, piRNAs in White Leghorn, and piRNAs in Red Jungle Fowl. Blue represents Watson-strand piRNAs; Red represents Crick-strand piRNAs. (**C**) Box plots showing abundance of piRNA precursors in 12 chicken tissues. (**D**) Venn diagram showing piRNA clusters defined in Red Jungle Fowl (White) and White Leghorn (Black). (**E**) Normalized RNA-seq reads of piRNA precursors in White Leghorn, White Leghorn piRNAs, unique mapping piRNAs, and Red Jungle Fowl small RNA reads (>23 nt).

The following figure supplements are available for figure 4:

**Figure supplement 1.** Divergent transcription of piRNA clusters.

**Figure supplement 2.** ALVE6 existed in chicken genome prior domestication.

Although the ALVE6 locus is defined as a piRNA cluster, it remains possible that ALVE piRNAs in White Leghorn are primarily derived from more recent insertions that occurred during domestication. Each ERV insertion event typically deposits a full-length provirus, as the ERVs are reverse transcribed to double-strand DNAs before their integration into the genome (*Lewinski and Bushman, 2005*). Additionally, an intact ALVE site cannot explain why uniquely mapping piRNAs come from the flanking genomic regions of ALVE6 (*Figure 4E*). Moreover, because ALVE6 has accumulated SNPs that are observed as uniquely mapping reads differentiating ALVE6 from other ALVEs (*Figure 4E*), among the piRNAs overlapped with the 33 SNPs that discriminate new ALVE insertions from ALVE6, 73.3% were expressed from ALVE6 locus and exhibited a pronounced 1st U bias, indicating that ALVE6 was the primary source for ALVE piRNAs. The piRNAs expressed from the new ALVE insertions exhibited a pronounced 10th A bias (*Figure 4—figure supplement 2B*), indicating that they represented secondary piRNAs generated during Ping-Pong amplification of the ALVE piRNAs. Finally, the presence of ALVE6 in chicken suppresses the spontaneous activation of intact ALVE copies, and enhanced fitness has been associated only with truncated ALVE and not with full length ALVE provirus (*Smith et al., 1990a*). Therefore, we conclude that the ALVE piRNAs are primarily produced from the pre-existing ALVE6 locus.

## Discussion

In this study, we found that a truncated ALVE provirus gave rise to the piRNAs that target ALVE in White Leghorn. ALVE6 had been identified as a dominant gene that confers resistance to the horizontal spread of spontaneously expressed ALVE (*Robinson et al., 1981*) and to congenital transmission of ALVE (*Smith et al., 1990a*). Although ALVE6 is a defective provirus, and is not infectious, it is highly expressed in domestic fowl (*Hayward et al., 1980*). The truncated provirus produces envelope glycoproteins, and it was proposed that products of ALVE6 compete for cellular receptors (*Robinson et al., 1981*), thus, preventing ALVE replication. However, the expression of envelope proteins from ALVE6 only leads to a 3–4 fold reduction in virus penetration, and does not account for robust resistance to ALVE infection in chickens (*Robinson et al., 1981*). More than 30 years ago, before the discovery of piRNAs, Robinson et al speculated that 'the presence of endogenous virus...would protect the germ line from accumulation of provirus and provirus-associated mutations' (*Robinson et al., 1981*). This type of 'immune response' is also known as viral interference—most hosts are resistant to infection by viruses expressed by their germ-line provirus. Although chicken piRNA mutant is currently not available to estimate the magnitude of restriction rendered by the piRNA pathways, in mouse mutant with disrupted PIWI gene, the transposon expression increased up to 10 fold (*Aravin et al., 2007*). Therefore, our discovery represents a new viral interference mechanism, and provides a critical missing piece to the puzzle: we attribute the ability to protect the chicken germ line, at least partially, to the function of piRNAs produced by truncated ALVEs.

The example presented here of piRNAs targeting an infectious virus in vertebrates represents a previously unappreciated function and history of piRNAs. Generally, there is a clear boundary between TEs and viruses. Most TEs are in a long-term co-evolutionary relationship that minimizes deleterious impacts on the host, but most viruses adopt a more destructive lifestyle that often leads to intense conflict with their hosts (*Feschotte and Gilbert, 2012*). In fruit flies, as a response to different parasites, anti-viral responses are mediated by endogenous siRNAs, and TE silencing is mediated by piRNAs; the functional division of the two small RNA pathways is clear—piRNAs do not appear to play a role in anti-viral defense (*Goic et al., 2013*). Vertebrate genomes, however, contain ERVs (*Malik et al., 2000*), which blurs the boundary between the TEs and viruses. Most ERVs, due to loss-of-function mutations, have lost the ability to make infectious viral particles. This loss occurs evolutionarily, and their infectious capacity is likely maintained for some period of time after endogenization. In addition to ALV in chickens, infectious ERVs have been reported in mammals, including mouse mammary tumor virus (MMTV) (*Moore et al., 1979*), Moloney Murine Leukemia Virus (M-MuLV) (*Stoye and Coffin, 1987*), Koala retrovirus (KoRV) (*Tarlinton et al., 2008*), and Feline ERV-DC (*Anai et al., 2012*). Acting as an essential immune response in germ-line defense, piRNAs would be expected to have evolved before ERVs lost their infectious capacity, in which case piRNAs would have contributed to host defense against the infectious viruses. Our results here expand the function of piRNAs to include resistance to infectious pathogens in vertebrates, and imply that the vertebrate

piRNA pathway has evolved under selection pressures both from the mutagenic effects of TE propagation and from the deleterious effects of activation of infectious ERVs.

The new piRNAs that we report were produced from an existing genomic element, rather than via 'trapping' a new ALVE insertion into a highly expressed piRNA cluster. The birth of new piRNA loci shown here is reminiscent of the origin of new genes from previously noncoding DNA (*Schlötterer, 2015*) that acquire additional regulatory signals for transcription and acquired a functional ORF. The transcription of ALVE6 in White Leghorn could be activated by an adjacent transcriptional promoter as proposed previously (*Tereba, 1981*), or by de novo acquisition of a transcriptional binding motif derived via point mutations. Our detection of ALVE6 expression in Red Jungle Fowl by RNA-seq indicates that transcription alone is not sufficient to become a new piRNA-producing locus. Considering that the genomic structure of ALVE6 is similar between White Leghorn and Red Jungle Fowl, either point mutations or epigenetic changes mark the ALVE6 transcripts for piRNA production in White Leghorn. Although we do not understand the mechanisms that lead to conversion of quiescent genomic regions to emerge as active piRNA producing loci, our work identifies a new mechanism for piRNA acquisition for ERV defense through 'twisting' existing elements.

piRNAs are the most recently discovered family of small silencing RNAs, and questions regarding the biogenesis and function of piRNAs remain. For example, a large proportion of non-TE piRNAs mysteriously enable mammalian sperm production (*Reuter et al., 2011*; *Lim et al., 2015*). Each of the available model organisms, including fruit fly, zebrafish and mouse, exhibits distinct piRNA features, and provides unique insights into piRNA pathway. Chicken piRNAs exhibit unique hybrid features of piRNAs found in other organisms. For example, the convergent transcription of piRNA-producing loci resembles the dual strands in fruit flies (*Brennecke et al., 2007*; *Malone et al., 2009*), but is distinct from that in frog (*Chirn et al., 2015*), zebrafish (*Houwing et al., 2007*), and mice (*Li et al., 2013*). However, unlike piRNAs in fruit flies and zebrafish that derive mainly from TEs, fewer than 10% of chicken piRNAs come from TE regions. The majority of chicken piRNAs come from non-TE regions, similar to piRNAs in adult mouse testis. Thus, the 1633 piRNA-producing loci should provide a valuable resource for the study of chicken piRNAs that will enable us to unify distinct features in model organisms.

Chicken breeding is based on quantitative traits. Putative new insertions, ERV activity, and the capacity to produce piRNAs can be potential contributors to the genetic changes that underlie phenotypic selection. We observed that 58 putative new ERVs inserted into protein coding genes in White Leghorn. Some of these were known to be associated with commercial traits; the others were mapped here for the first time. Each insertion is a mutational event, and has the potential for altering the phenotype. These insertions may contribute to the individual variations of chickens with respect to growth rate, egg production, woody meat, response to heat stress, and resistance to pathogens including newcastle disease virus, avian influenza virus, clostridium, campylobacter, and salmonell. All new insertions are derived from young TEs that are controlled by piRNAs encoded within piRNA-producing loci. Considering the intra-species diversity of piRNA producing loci, both repressive and non-repressive alleles may exist. During selective breeding, it is possible that a genomic region responsible for TE piRNA production is segregated from the active TEs, resulting in TE activation in germ line of F1 generation and increased TE insertions in the offspring of F1. High ALVE levels and increased integrations have been associated with low body weight in chickens (*Ka et al., 2009a*, *2009b*). Therefore, the identified active TEs and piRNA clusters may provide another angle for the discovery of functional polymorphisms underlying quantitative traits, and may also be used to guide breeding to modulate TE activity.

Two observations in our studies indicated the presence of germ-line TE control mechanisms beyond piRNAs. To wit, intact ALVE is present and expressed in Red Jungle Fowl, and upon induction, it can produce infectious viral particles (*Weiss and Biggs, 1972*). Despite this, and despite the absence of ALVE piRNAs, variations of genomic copy numbers of ALVE in Red Jungle Fowl have not been reported. It could be that the somatic TE suppression mechanisms, such as histone modification and DNA methylation, are sufficient to protect Red Jungle Fowl from re-integration. Alternatively, activation, when it occurs, may be extremely deleterious, preventing spread to the general population (*Lu and Clark, 2010*). A second mechanism is suggested by the observation that no piRNA-producing loci or PIWI genes are expressed in the ovary, a site where ERVs are highly expressed. In mice, another small RNA silencing mechanism mediated by endogenous siRNAs is known to protect the murine oocytes from TE attacks (*Tam et al., 2008*; *Watanabe et al., 2008*). As

a consequence, piRNA pathways are not essential for female mouse fertility (*Kuramochi-Miyagawa et al., 2004*; *Carmell et al., 2007*). Although the activation of siRNAs in oocytes is rodent-specific, suggesting that this piRNA-independent defense may not exist in other mammals (*Flemr et al., 2013*; *Rosenkranz et al., 2015*), our studies suggest that sexual dimorphism in piRNA pathway may be a conserved feature.

In conclusion, chicken piRNAs have rapidly evolved to protect the germ-line genome from the contemporary threats. The robust Ping-Pong amplification in piRNAs targeting young TEs reflects an ongoing arms race. When TEs become inactive, the piRNAs gradually erode away, as shown by the low abundance of medium TE piRNAs and the death of piRNAs targeting old TEs. A mystery surrounds the means by which new piRNAs are acquired when a retrovirus is endogenized to a new host. The compact chicken genome, which includes a small fraction of TEs (10%) (*International Chicken Genome Sequencing Consortium, 2004*), permitted pinpointing the ALVE piRNA-producing locus and tracing its evolutionary history. In chickens, ALVE6, as a piRNA-producing locus, exhibits 'on' and 'off' states. Comparative studies among chicken breeds will delineate the molecular events that turn on piRNA production at the ALVE6 locus. The acquisition of piRNA that target a recently invaded ERV, as reported here, represents an opportunity to elucidate the mechanisms by why some transcripts produce piRNAs while some do not.

## Materials and methods

### Animals
Rooster testes from a 15 months-old White Leghorn of the Cornell Special C strain were used for polysome gradients, ribosomal profiling, RNA-seq, and genomic PCR. The same strain was used to construct the small RNA libraries in our previous studies (*Li et al., 2013*).

### Polysome profiling
Testes were flash frozen in liquid nitrogen, and lysed in 1 ml lysis buffer (10 mM Tris-HCl, pH 7.5, 5 mM MgCl$_2$, 100 mM KCl, 1% Triton X-100, 2 mM DTT, 100 µg/ml cycloheximide, and 1× Protease-Inhibitor Cocktail). Lysates were homogenized with a pellet pestle for a total of ten strokes, and incubated at 4°C with inverted rotation for 10 min. The lysates were centrifuged at 1300 $g$ for 10 min at 4°C, the supernatant was recovered, and the absorbance at 260 nm was measured. Five A$_{260}$ absorbance units were used for polysome gradients and ribosome profiling.

Samples were loaded on a 10–50% (w/v) linear sucrose gradient (20 mM HEPES-KOH, pH 7.4, 5 mM MgCl$_2$, 100 mM KCl, 2 mM DTT, 100 µg/ml of cycloheximide) and centrifuged in a SW-40ti rotor at 35,000 rpm for 2 hr 40 min at 4°C. Samples were then collected from the top of the gradient using the gradient Fractionation system (BR-188, Brandel, Boca Raton, FL, USA) while monitoring absorbance at 254 nm was measured.

Synthetic spike-in RNAs were added to each collected fraction before RNA purification. The collected fractions were incubated at 42°C in 1% SDS and proteinase K (200 µg/ml) for 45 min. After proteinase K treatment, RNAs were extracted with one volume of Acid phenol (pH 4.5)/chloroform/isoamyl alcohol (25:24:1). The recovered aqueous phase was supplemented with 20 µg glycogen and precipitated with three volumes of 100% ethanol at 4°C for 1 hr. Pellets were washed with 70% ethanol, and RNAs were resuspended in water.

### Ribosome profiling
Ribosome profiling was performed as described (*Guo et al., 2010*; *Ingolia et al., 2012*; *Ricci et al., 2014*; *Cenik et al., 2015*) with the following modifications: Cleared testis lysates were incubated with 60 units of RNase T1 (Fermentas, Waltham, MA, USA) and 100 ng of RNase A (Ambion, Waltham, MA, USA) per A$_{260}$ unit for 30 min at room temperature. Samples were loaded on a 10–50% (w/v) linear sucrose gradient, and after centrifuged, the fractions corresponding to 80S monosomes were recovered.

Ribosome profiling Illumina-compatible sequencing libraries were prepared as follows (*Figure 1—figure supplement 3A*): (i) the RPFs were resolved on a 15% acrylamide (19:1) 8 M urea denaturing gel for 1 hr 30 min at 35 W, and fragments ranging from 26 nt to 35 nt were size-selected from the gel; (ii) size-selected RNAs were extracted from the gel slice by electro elution using GeBAflex tubes

(Gerad Biotech, Oxford, OH, USA), and the rRNAs were removed by Ribo-Zero Gold (Epicentre Bio-technologies, Madison, WI, USA); (iii) the 3′ ends of recovered RNAs were dephosphorylated by T4 PNK (New England BioLabs, Ipswich, MA, USA) in MES buffer (100 mM MES-NaOH pH 5.5, 10 mM MgCl$_2$, 10 mM $\beta$-mercaptoethanol and 300 mM NaCl) at 37°C for 3 hr, followed by Alkaline Phosphatase (New England BioLabs) treatment at 37°C for 1 hr; (iv) dephosphorylated RNAs were used in our small RNA library construction protocol with an additional step of 5′ end phosphorylation by T4 PNK (New England BioLabs) using the PNK buffer with 1 mM ATP at 37°C for 1 hr before 5′ ligation.

## RNA-seq

Strand-specific RNA-seq libraries were constructed following the TruSeq RNA sample preparation protocol as previously described (*Li et al., 2013*). rRNAs were depleted from total RNAs by Ribo-Zero Gold (Epicentre Biotechnologies, Madison, WI, USA). The library was sequenced using the paired-end 2 × 50 nt platform on a HiSeq 2000.

## Quantitative real-time PCR (qRT–PCR)

Extracted RNAs were treated with Turbo DNase (Thermo Fisher, Waltham, MA, USA) for 20 mins at 37°C and then size-selected to isolate RNA $\geq$200 nt (DNA Clean and Concentrator−5, ZYMO RESEARCH, USA) before reverse transcription by SuperScript III (Life Technologies, Carlsbad, CA, USA) at 50°C. Quantitative PCR (qPCR) was performed using the ABI Real-Time PCR Detection System with SYBR Green qPCR Master Mix (Bimake, Houston, TX, USA). Data were analyzed using DART-PCR (*Peirson et al., 2003*). Spike-in RNA was used to normalize RNAs in different fractions. *Supplementary file 1* lists the qPCR primers.

## Phylogenetic tree

PIWI protein sequences were obtained from the Ensembl genome browser (SCR_013367). Multiple sequence alignment and neighbour-joining clustering were performed with *clustalw* 2.0.12 (*Thompson et al., 1994*). The R package *ape* (*Paradis et al., 2004*) was used to create the phylogenetic tree.

## TE families

We used 200 chicken TE families that are defined in both Repbase (*Jurka et al., 2005*) and RepeatMasker (*Smit et al., 2016*, SCR_012954). We downloaded the 233 *Gallus gallus* and ancestral (shared) repeats from Repbase, and first removed the 46 families containing tRNAs, rRNAs, and snRNAs. Because Repbase and RepeatMasker sometimes name TEs differently, we submitted the Repbase repeat sequences to CENSOR (*Kohany et al., 2006*) or to blast to identify the corresponding RepeatMasker name. In comparing the TE annotation between RepeatMasker and Repbase, we found that 9 Repbase repeats appeared to be truncations of existing repeats. For example, CAM1_GG appeared to be an incomplete sequence of CR1-C4. Based on the latest chicken genome assembly (Gallus_gallus-5.0), we further removed 12 Repbase repeats did not have corresponding genomic copies. We also noticed that some TEs annotated in the genome by RepeatMasker were not included in the *Gallus gallus* repeats in Repbase. One example is EAV-HP, which is deposited in the archive Repbase21.08, but is classified as being of virus origin rather than chicken origin. We extracted the 34 repeats that are annotated in the current chicken genome by RepeatMasker from the vertebrate archive Repbase. The final total set of 200 TE families and their corresponding names in Repbase and Repeatmasker are listed in *Supplementary file 1*.

## General bioinformatics analyses

Analyses were performed using piPipes v1.4 (*Han et al., 2014*). All data from the small RNA-seq, ribosome profiling, RNA-seq, and genome sequencing were analyzed using the latest chicken genome release galGal5 (GCA_000002315.3). Generally, one mismatch is allowed for genome mapping and three mismatches are allowed for transcriptome mapping. For small RNA analysis, the transcriptome included the 200 TE families and 1633 piRNA clusters. For RNA-seq and ribosome profiling analysis, the transcriptome included mRNAs, lncRNAs, piRNA clusters, and TE families.

*Supplementary file 1* reports the statistics for the high-throughput sequencing libraries constructed in this study.

In small RNA-seq analysis, reads were mapped to ALVE and EAV-HP sequences before being mapped to the genome, and three mismatches were allowed for alignment. The sequences of EAV-HP and ALVE came from NCBI with id: NC_005947.1 (*Sacco et al., 2000*) and AY013303 (*Johnson and Heneine, 2001*). We analyzed previously published testis small RNA libraries from White Leghorn (GSM1096613) (*Li et al., 2013*), and from Red Jungle fowl (GSM995329) (*Meunier et al., 2013*). Small RNA species with characteristic piRNA length (>23 nt) were defined as piRNAs (*Ghildiyal et al., 2008*). The small RNA libraries from White Leghorn and Red Jungle Fowl were normalized to the sum of all piRNA reads. Oxidized samples were calibrated to the corresponding total small RNA library via the abundance of shared piRNA species. The piRNA abundance per TE or per piRNA cluster is reported either as parts per million reads mapped to the genome (ppm) or reads per kilobase pair per million reads mapped to the genome (rpkm) using a pseudo count of 0.01.

The pair-end total RNA-seq reads were aligned to the genome using TopHat 2.0.12 (*Trapnell et al., 2009*, SCR_013035). Reads were mapped using the '-g 100' flag. The direct transcriptome mapping results were quantified using eXpress (*Roberts and Pachter, 2013*, SCR_006873). The advantage of eXpress lies in the Expectation–Maximization algorithm to apportion multimapping reads, reporting the estimated numbers of fragments in each transcript (*Dempster et al., 1977*). The eXpress results are normalized by the gene compatible reads calculated by Cufflinks per library; and the fpkm (fragments per kilobase of transcript per million mapped reads) value with a pseudo count of 0.01 was used for all analyses. We analyzed our RNA-seq library from testis of White Leghorn and the published RNA-seq libraries from different tissues of Red Jungle Fowl (GSM752557, GSM752558, GSM752559, GSM752560, GSM752561, GSM752562, GSM752563, GSM752564, GSM752565, GSM752566, GSM752567, GSM752568, GSM1064853, GSM1064854, GSM1064855, and GSM1196055) (*Brawand et al., 2011*; *Necsulea et al., 2014*).

Ribosome profiling analysis was done according to the modified small RNA pipeline procedure, but including the junction mapping reads. Ribosome protected fragments (RPFs) 26–32 nt long were selected for further analysis. The RPF abundance per TE or per piRNA cluster was quantified by eXpress, and reported as reads per kilobase pair per million reads mapped to the genome (rpkm) using a pseudo count of 0.01.

The pair-end genome sequencing reads were aligned to the reference genome using BWA-aln (-R 1000) (*Li and Durbin, 2009*, SCR_010910). We analyzed the previously published genome resequencing libraries from White Leghorn (SRX1121834, SRX1121835, SRX1121836) (*Oh et al., 2016*) and we combined the three replicates to increase detection sensitivity. The new transposition events were analyzed by TEMP (*Khurana et al., 2011*; *Zhuang et al., 2014*, SCR_001788). The insertions that are supported by reads at both sides are listed in *Supplementary file 1*.

Statistical analyses were performed in R 3.0.2 (*Team, 2014*, SCR_001905). The significance of the differences was calculated by Wilcoxon rank sum test except as indicated in the text.

## Ping-Pong analysis

Ping-Pong amplification was analyzed by the 5′–5′ overlap between piRNA pairs from opposite genomic strands (*Li et al., 2009*). Overlap scores for each overlapping pair were the product of the number of reads of each of the piRNAs from opposite strands. The overall score for each overlap extension (1–30) was the sum of all such products for all chromosomes. Heterogeneity at the 3′ ends of small RNAs was neglected. The Z-score for a 10 bp overlap was calculated using the scores of overlaps from 1–9 and 11–30 as background.

## Nucleotide periodicity

Nucleotide periodicity was computed as described (*Pelechano et al., 2015*) with modifications. We first aligned the RPFs to each other using 5′–5′ overlap analysis from the same transcript, and reported the distance spectrum. An annotated ORF is not a prerequisite for this analysis. The distance spectrum of RPFs from mRNAs already showed a 3-nt periodicity pattern. We then transformed the distance spectrum using the 'periodogram' function of the *GeneCycle* package

(*Wichert et al., 2004*) with the 'clone' method. The relative spectral density was calculated by normalizing to the value at the third position.

## Rooster piRNA-producing loci detection

We used the same dynamic programming algorithm that we developed previously (*Li et al., 2013*) to identify genomic regions with the highest piRNA density. The oxidized small RNA reads (>23 nt) (SRR772069) were used to define the clusters in White Leghorn, and the small RNA reads (>23 nt) (SRR553601) were used to define the clusters in Red Jungle Fowl. We assumed that piRNA clusters comprise at most 5% of the chicken genome. We first split the genome into one kbp non-overlapping windows, and computed piRNA abundance for each window. The mean of the top 5% of windows was used as the penalty score for the dynamic programming algorithm. The algorithm computes the cumulative piRNA abundance score as a function of the window index along each chromosome. The score at a window is the sum of: the score in the previous window, plus the piRNA abundance in the current window, minus the penalty score; negative scores were reset to 0. The maximum score points to the largest piRNA cluster. We extracted the largest piRNA cluster, recomputed the scores at the corresponding windows, and searched for the next cluster. This process was continued iteratively until the scores for all windows were zero. The boundaries of each cluster were further refined by including those base pairs for which piRNA abundance exceeded the mean piRNA abundance of the top 5% windows. We required a piRNA cluster to have at least one unique mapping read. The coordinates of all 1633 piRNA-producing loci of White Leghorn and whether they are conserved in Red Jungle Fowl are reported in *Supplementary file 1*.

## Rooster testis transcriptome annotation

We used Cufflinks v2.2.1 (*Trapnell et al., 2012*, SCR_014597) with parameters of '-u -j 0.2 –min-frags-per-transfrag 40 –overlap-radius 100' to assemble transcripts using the strand specific pair-end RNA-seq data from adult testis of White Leghorn (*Supplementary file 1*). We assembled 59,614 transcripts. Using the TransDecoder/3.3.0 (*Haas et al., 2013*) with the BlastP (retain ORFs with homology to known proteins) and Pfam search (identify common protein domains), we further identified the candidate coding regions of 49,962 mRNA transcripts. We performed our transcriptome analysis on 9505 mRNAs which had an abundance of at least 10 fpkm in testis: among these mRNAs, 9287 were reported in the latest release of RefSeq (GCF_000002315.4, SCR_003496), and 218 were putative novel mRNAs. For lncRNAs, we first selected the 13,103 assembled transcripts that were reported to be lncRNA by RefSeq. Among the 724 lncRNAs with an abundance above 10 fpkm in testis, 218 of these transcripts had ORFs detected by TransDecoder. Following removal these putative false lncRNA, we performed transcriptome analysis on the remaining 502 lncRNAs.

## Defining chicken Malat1 sequence using INFERNAL and covariance model

We created a covariance model (C*ƒ*M) using Infernal v1.1rc1 as previously described (*Nawrocki and Eddy, 2013*, SCR_011809). Briefly, we built a CM based on the human *mascRNA* and *menRNA* alignment using the cmbuild program, which was calibrated for E-value reporting with the cmcalibrate program. We then searched the chicken genome for high scoring hits with the cmsearch program. Default Infernal v1.1rc1 parameters were used for all steps (cmbuild, cmcalibrate, and cmsearch programs). Only one significant hit with an E value below 0.01 was identified by the CM model. This tRNA-like element is from the minus strand of chrUn_AADN03016580:1547–1491 with the E value of $3.8 \times 10^{-9}$. Manual inspection of its upstream sequence revealed a MALAT 3′ end like module with two T-rich motifs: TTTTCTTTT and TTTTGCTTTT, and one polyA-like moiety: AAAAAAAGCAAAA. This contig contains 6473 bp and does not harbor TEs. While ESTs mapped of this tRNA-line element, hundreds ESTs mapped to sites spanning the entire upstream region (chrUn_AADN03016580:1492–6473), suggesting that the promoter of this gene lies outside of this contig. Despite the lack of syntenic information, it has been shown to be a human MALAT1 homolog in chicken. The evolution of MALAT1 lncRNA and its 3′end module is reported in a manuscript under review (*Zhang et al., 2017*).

## Data access

All sequence data reported here are available through the NCBI Gene Expression Omnibus under the accession number GSE93559.

## Acknowledgements

We thank R Okimoto, T Eickbush, and A Larracuente for discussions; P Johnson for providing rooster testes; L.Huang for providing key references; G Ansah, K Boundy, C Roy, L Maquat, R Viswanatha, Z Zhang, A McDavid, and members of the Li laboratory for advice and critical comments on the manuscript. This work was supported in part by National Institutes of Health grants R00HD078482 to XZL.

## Additional information

### Funding

| Funder | Grant reference number | Author |
| --- | --- | --- |
| National Institutes of Health | R00HD078482 | Xin Zhiguo Li |

The funders had no role in study design, data collection and interpretation, or the decision to submit the work for publication.

### Author contributions

YHS, Conceptualization, Resources, Data curation, Software, Formal analysis, Supervision, Funding acquisition, Validation, Investigation, Visualization, Methodology, Writing—original draft, Project administration, Writing—review and editing; LHX, Data curation, Software, Formal analysis, Writing—review and editing; XZ, Data curation, Investigation, Methodology, Writing—review and editing; QC, Conceptualization, Investigation, Writing—original draft, Writing—review and editing; DG, Data curation, Investigation, Writing—review and editing; BZ, Software, Formal analysis, Writing—review and editing; JJ, XZL, Conceptualization, Resources, Writing—review and editing; CY, Resources, Supervision, Writing—review and editing

### Author ORCIDs

Yu Huining Sun, http://orcid.org/0000-0003-0333-2898
Xiaoyu Zhuo, http://orcid.org/0000-0002-1400-5609
Xin Zhiguo Li, http://orcid.org/0000-0001-6803-4600

## Additional files

### Supplementary files

• Supplementary file 1. Detailed information and statistics for the sequencing data used in this study. (A) Ribosome profiling sequencing statistics: reads and species. (B) Small RNA sequencing statistics: reads and species. (C) RNA-Seq statistics: reads and species. (D) 200 TE families. (E) TE insertions defined by TEMP. (F) Genome coordinates for the 1633 rooster piRNA-producing loci defined in this study are provided in UCSC BED format (i.e., 0-based) for galGal5. (G) Primers used in this study for qRT-PCR and genomic PCR.

### Major datasets

The following dataset was generated:

| Author(s) | Year | Dataset title | Dataset URL | Database, license, and accessibility information |
| --- | --- | --- | --- | --- |
| Yu Huining Sun, Xin Zhiguo Li | 2017 | Domestic chickens activate a piRNA defense againstavian leukosis virus | https://www.ncbi.nlm.nih.gov/geo/query/acc.cgi?acc=GSE93559 | Publicly available at NCBI Gene Expression Omnibus (accession no: GSE93559) |

The following previously published datasets were used:

| Author(s) | Year | Dataset title | Dataset URL | Database, license, and accessibility information |
|---|---|---|---|---|
| Li XZ, Roy CK, Zamore PD | 2013 | An Ancient Transcription Factor Initiates the Burst of piRNA Production During Early Meiosis in Mouse Testes | https://www.ncbi.nlm.nih.gov/geo/query/acc.cgi?acc=GSE45049 | Publicly available at NCBI Gene Expression Omnibus (accession no: GSE45049) |
| Meunier J, Lemoine F, Soumillon M, Liechti A, Weier M, Guschanski K, Hu H, Khaitovich P, Kaessmann H | 2013 | Evolution of mammalian miRNA genes | https://www.ncbi.nlm.nih.gov/geo/query/acc.cgi?acc=GSE40499 | Publicly available at NCBI Gene Expression Omnibus (accession no: GSE40499) |
| Brawand D, Soumillon M, Necsulea A, Julien P, Csárdi G, Harrigan P, Weier M, Liechti A, Aximu-Petri A, Kircher M, Albert FW, Zeller U, Khaitovich P, Grützner F, Bergmann S, Nielsen R, Pääbo S, Kaessmann H | 2011 | The evolution of gene expression levels in mammalian organs | https://www.ncbi.nlm.nih.gov/geo/query/acc.cgi?acc=GSE30352 | Publicly available at NCBI Gene Expression Omnibus (accession no: GSE30352) |
| Necsulea A, Soumillon M, Liechti A, Daish T, Zeller U, Baker J, Grutzner F, Kaessmann H, Warnefors M | 2014 | The evolution of lncRNA repertoires and expression patterns in tetrapods | https://www.ncbi.nlm.nih.gov/geo/query/acc.cgi?acc=GSE43520 | Publicly available at NCBI Gene Expression Omnibus (accession no: GSE43520) |
| Oh D, Son B, Mun S, Oh MH, Oh S, Ha J, Yi J, Lee S, Han K | 2016 | Whole genome resequencing for 3 different domesticated chicken breeds (White Leghorn, Korea domestic and Araucana) | https://trace.ncbi.nlm.nih.gov/Traces/sra/?study=SRP061672 | Publicly available at NCBI Sequence Read Archive (accession no: SRP061672) |

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
