## [Decision Letter]

Thank you for submitting your article "Domestic chickens activate a piRNA defense against avian leukosis virus" for consideration by *eLife*. Your article has been favorably evaluated by James Manley (Senior Editor) and three reviewers, one of whom, Stephen P Goff (Reviewer #1), is a member of our Board of Reviewing Editors. The following individual involved in review of your submission has agreed to reveal their identity: Karen L Beemon (Reviewer #2).

The reviewers have discussed the reviews with one another and the Reviewing Editor has drafted this decision to help you prepare a revised submission.

We send along here all three reviews. There are some specific issues that should be addressed in a revised draft. The reviewers are quite supportive and it appears that only a few changes in the text will suffice to produce an acceptable version of your manuscript.

*Reviewer #1:*

This paper provides a substantial body of data about piRNA function in avian species and offers a compelling argument for the role of piRNAs in control of both ERVs and ALVs in domestic chickens. Most exciting is the proposal that a particular ERV (ALVE6) is the source of the piRNAs that target its family members, and that piRNAs may be the major mechanism used in the known antiviral activity of this locus. The model suggests that transcriptionally activating existing ERVs may be all that is needed to generate new sets of piRNAs.

The data include assays for RNA expression, but also for translation (both ribosome association and ribosome protection) across tissues and with comparisons between closely related wild and domestic species. There is much consideration of the evolutionary timing of appearance of ERVs and piRNA protection. There is a deep scan of piRNA content. The writing is clear, the review of the history (including that of piRNAs and ERVs in many species) is extensive, and the conclusions are appropriately voiced. I found the paper exciting.

*Reviewer #2:*

This is a very interesting paper and the work seems to have been carefully done.

They should add the following reference, which first showed chicken ERVs were transcribed and translated:

Unexpected diversity and expression of avian endogenous retroviruses.

Bolisetty M, Blomberg J, Benachenhou F, Sperber G, Beemon K.

MBio. 2012 Oct 16;3(5):e00344-12. doi: 10.1128/mBio.00344-12.

*Reviewer #3:*

This paper describes an investigation of piRNA activity in chickens. It is an impressive piece of work and I think the findings are potentially highly impactful. The manuscript describes the discovery that, in chickens, piRNA defenses have evolved in relatively shallow evolutionary time against a lineage of retroviruses (avian leukosis virus [ALV]) that exists both as an exogenous virus and as endogenous loci in the chicken genome. Remarkably, the authors observed that in domestic chickens – in which ALV is an important pathogen – one ALV locus (ALVE6) is part of a piRNA cluster, while the same locus does not produce piRNAs in the red jungle fowl, the chicken's wild ancestor. This finding would seem to indicate that piRNA production from specific loci exists in an 'on/off' state, can target infectious viruses, and can adapt relatively rapidly to provide germline defense against newly acquired transposable elements. The chicken provides an excellent system in which to investigate this phenomenon, as the genome has a relatively low TE content, and chicken retroviruses (both exogenous and endogenous) have been extensively studied. The paper raises interesting questions about piRNA-based defense against retroviruses in the vertebrate germline. The text makes clear where there are possible alternative interpretations of the data, and about the open questions remaining with respect to the phenomenon of piRNA defense in vertebrates.

The paper is well-written overall, although in places some minor edits might be helpful for the purposes of clarity. I would think it important that the paper is reviewed by someone experienced in using the polysome and ribosome profiling techniques, and piRNA-associated bioinformatics tools that are applied here (which I am not) in case there is any possibility of misleading artefacts, but as far as I was able to assess the experiments have been performed appropriately and the data seem robust.

I have some questions about the way the authors grouped TEs as young or old. Was a TE defined as old because the lineage could be shown to have been present in the germline for a long time? On this basis, murine ERV-L (MuERV-L) would be defined as ancient – which is in a way correct because ERV-L entered the mammalian germline at least 100 million years ago. But MuERV-L has also been active relatively recently in murids, and most copies in the mouse genome are relatively young, so I am curious as to how this kind of element would have been categorised in the approach the authors describe.

This is a relatively minor criticism, and I would not insist on a change, but I am not sure the authors have selected the best set of figures to illustrate their findings. I understand why each panel is included in the paper as a whole. Not sure all of the panels need to be in the main text – Figure 1 and Figure 1 in particularly don't seem particularly helpful to conveying the main message of the paper, unless I have missed something.

---

## [Author Response]

*[…] Reviewer #2:*

*This is a very interesting paper and the work seems to have been carefully done.*

*They should add the following reference, which first showed chicken ERVs were transcribed and translated:*

Unexpected diversity and expression of avian endogenous retroviruses.

Bolisetty M, Blomberg J, Benachenhou F, Sperber G, Beemon K.

*MBio. 2012 Oct 16;3(5):e00344-12. doi: 10.1128/mBio.00344-12.*

Thank you for pointing out, we added the reference, please see the first paragraph of the subsection “Identifying active TEs in domestic fowl”.

*Reviewer #3:*

*[…] I have some questions about the way the authors grouped TEs as young or old. Was a TE defined as old because the lineage could be shown to have been present in the germline for a long time? On this basis, murine ERV-L (MuERV-L) would be defined as ancient – which is in a way correct because ERV-L entered the mammalian germline at least 100 million years ago. But MuERV-L has also been active relatively recently in murids, and most copies in the mouse genome are relatively young, so I am curious as to how this kind of element would have been categorised in the approach the authors describe.*

In Figure 2, we collected the age information of all 200 TE families in the chicken genome from Repbase. Specifically, the OS (Organism Species) and OC (Organism Classification) lines in their EMBL sequence format. In the case of MuERV-L, the OS in Repbase annotation is *Mus musculus* (http://www.girinst.org/protected/repbase_extract.php?access=MERVL&form at=EMBL). The OS indicates that they are specific to mice, consistent with their recent activity as pointing out by reviewers. Therefore, in our initial draft, we used Repbase OS annotation instead of just the entrance age of the TE family. We assumed that the OS in Repbase is based on TE invasion age, but it is not always the case as pointed out by the reviewer. To accurately describe our information source used in Figure 2, we replaced the sentence “Given the rare occurrence of horizontal transfer among vertebrates, the time when the TE family first entered the vertebrate genome has been inferred from their host range” with “we inferred TE age using organism information available in Repbase”.

*This is a relatively minor criticism, and I would not insist on a change, but I am not sure the authors have selected the best set of figures to illustrate their findings. I understand why each panel is included in the paper as a whole. Not sure all of the panels need to be in the main text – Figure 1 and Figure 1 in particularly don't seem particularly helpful to conveying the main message of the paper, unless I have missed something.*

We agree that the panels in Figure 1 are somewhat redundant, and we have removed the qPCR results of CR1B ORF1 and CR1F ORF1. For Figure 1, from the outside to inside, the 3^rd^ circle represents the insertion sites (subsection “Identifying active TEs in domestic fowl”, last paragraph), the 2^nd^ circle represents the piRNA cluster locations (subsection “Defining piRNA-producing loci in chickens”, first paragraph), and the 1st circle represents the position of centromeres on each chromosome.